# Fisheries portfolio diversification and turnover buffer Alaskan fishing communities from abrupt resource and market changes

Timothy J. Cline[1], Daniel E. Schindler[1] & Ray Hilborn[1]

Abrupt shifts in natural resources and their markets are a ubiquitous challenge to human communities. Building resilient social-ecological systems requires approaches that are robust to uncertainty and to regime shifts. Harvesting diverse portfolios of natural resources and adapting portfolios in response to change could stabilize economies reliant on natural resources and their markets, both of which are prone to unpredictable shifts. Here we use fisheries catch and revenue data from Alaskan fishing communities over 34 years to test whether diversification and turnover in the composition of fishing opportunities increased economic stability during major ocean and market regime shifts in 1989. More than 85% of communities show reduced fishing revenues following these regime shifts. However, communities with the highest portfolio diversity and those that could opportunistically shift the composition of resources they harvest, experienced negligible or even positive changes in revenue. Maintaining diversity in economic opportunities and enabling turnover facilitates sustainability of communities reliant on renewable resources facing uncertain futures.

[1] School of Aquatic and Fishery Sciences, University of Washington, 1122 NE Boat Street, Seattle, Washington 98105, USA. Correspondence and requests for materials should be addressed to T.J.C. (email: tjcline@uw.edu).

Ecosystem dynamics are notoriously unpredictable, especially considering the effects of ongoing climate change, ocean acidification and other poorly understood and interacting perturbations. It is often argued that more mechanistic science will lead to greater understanding of ecosystem functioning and improved forecasting for variability and change[1]. As such, there has been tremendous scientific effort to forecast and anticipate approaching regime shifts[2], yet the utility of these approaches is often challenged in real-world applications[3]. Our ability to accurately predict the timing and character of changes in ecosystems and populations will always be limited; thus, there is a distinct need to develop strategies to promote ecological and socio-economic resilience without reliance on accurate long-term forecasts[4].

A large proportion of the world's population lives in coastal communities and is critically dependent on highly volatile marine resources[5]. Fisheries are highly variable and risky as an income source[6] as the populations they depend on fluctuate strongly at interannual and interdecadal time scales[7,8]. Additionally, fisheries are coupled social-ecological systems that are affected not only by biological responses to oceanographic changes, but also by socio-economic conditions and market demand expressed at local to global scales[9]. For example, prices paid to fishermen can be highly volatile and may easily double or halve from year to year, depending on global market conditions[10]. Communities that rely on fisheries, or most natural renewable resources, for their livelihoods need to integrate over variability, shocks and reorganization of the integrated social-ecological system to sustain their economies and livelihoods[11]. The capacity of coupled human-natural systems to reorganize through turnover in composition while maintaining key functions in response to a changing environment is a hallmark of the resilience concept[12–14]. There is a pressing need to understand what factors enable resilience and potential for adaptation in social-ecological systems against new climate realities and globalized economies[15].

Forecasting and insurance are two strategies that have been used extensively to help buffer people against unexpected perturbations to natural resource economies; the former is used to avoid risks while the latter is used to transfer damages from negative perturbations[16]. In financial markets, investors use portfolio diversification to buffer against risk when forecasts have little power[17]. Investors also use turnover, the buying and selling of different assets, to adapt to shifting markets and exploit emerging opportunities. These concepts have a rich history in economic and biodiversity theory[18] and the range of benefits continues to become apparent[19]. Portfolio diversification, in terms of exploiting multiple species or populations, has been shown to stabilize interannual variability in commercial fisheries and fishing communities[6,11,20–22]. However, it remains unclear whether diversification or adaptation through turnover of opportunities[23] can provide resilience for communities in the face of large and abrupt regime shifts.

Abrupt shifts in ecosystems and fish stocks are common in the marine environment[7,8]. Importantly, these regime shifts can have major economic consequences as biological communities reorganize in response to ecosystem and ocean dynamics[7,24]. Regime shifts are also present in markets for natural resources[10,25,26]. For example, Pacific salmon have shown abrupt and persistent changes in their market price due to fluctuations in global aquaculture production of salmon[10]. Diversification (the distribution of participation across a number of fisheries), and turnover (reorganization of fishing effort among fisheries to capitalize on emerging opportunities), should alleviate economic hardships and provide resilience in the face of large and abrupt regime shifts. While the concepts of diversification and turnover have been considered key components of social-ecological resilience in a theoretical sense[13,14] and in small-scale fisheries[23], the magnitude of their effects on the resilience of social-ecological systems against regime shifts has yet to be quantified at a large scale with empirical data.

Here we use a well-documented oceanographic regime shift with far reaching biological consequences in the North Pacific Ocean[24,27], and a global market regime shift in the price of salmon[10], to test the hypothesis that diversifying fishing opportunities and enabling turnover in the composition of what is harvested provide resilience against abrupt shifts for fishery-dependent communities. We used data from Alaska where many communities are intimately tied to fishing for their livelihoods and where commercial catches and revenues have been fully monitored over three decades. We focus at the scale of communities because many of the socio-economic benefits of resilience are expressed at the community scale. For example, many Alaskan communities derive much of their tax revenue from fishery landings, fishing activity generates other commerce within the community, and their social identity is often expressed at the community scale[28]. Additionally, there exists a natural gradient in diversification among communities driven by geographic isolation and economic opportunity that provides contrast for comparison. These data comprise more than 60 fisheries, 100 individual communities and thousands of permit holders all susceptible to abrupt shifts in the resources and markets.

Our results show that diversification and turnover of fishing opportunities significantly reduce the negative impacts of regime shifts on the revenues generated by fishing communities. We find that >85% of fishing communities saw large decreases in their revenues following major ocean regime and market shifts. However, communities that participated in a more diverse collection of fisheries saw smaller changes in revenue and had more opportunities to adapt to the changes. In some cases, communities actually increased their revenue from fishing as they were able to capitalize on emerging opportunities. Enabling communities to create and maintain flexibility in fishing opportunities appears to be a tangible mechanism for providing resilience against unpredictable changes in ecosystems and markets. Building sustainable communities will require employing strategies that are robust to irreducible uncertainties about future changes in coupled human-natural systems.

## Results

**Ocean regime shift reorganizes commercial fisheries catch.** The ecosystems of the North Pacific Ocean are notoriously variable, expressing change at interannual and interdecadal time scales large enough to seriously challenge the sustainability of fishery-dependent communities. These unpredictable changes in productivity and species composition are due mostly to oceanographic changes that favour certain stocks or species. Importantly, these changes do not result in the unilateral collapse or benefit of all species; rather they cause a reshuffling of the relative abundances of important commercial species[24]. It has been well documented that, in 1989, there were major changes in productivity and species composition across the North Pacific that have persisted since the regime shift[24] (Fig. 1a). There were observed changes in recruitment of many species[24,27]. Oceanographic changes are most likely to impact juvenile fishes and recruitment[24,27]; therefore, the timing of changes in adult stock size and impacts on fisheries will be dependent on the life-history of the species and age they recruit to the fishery[29].

To understand how the 1989 regime shift affected commercial fisheries across Alaska, we evaluated changes over time in commercial fisheries harvest by stock. Using dynamic factor analysis (DFA)[30], a dimension reduction technique developed

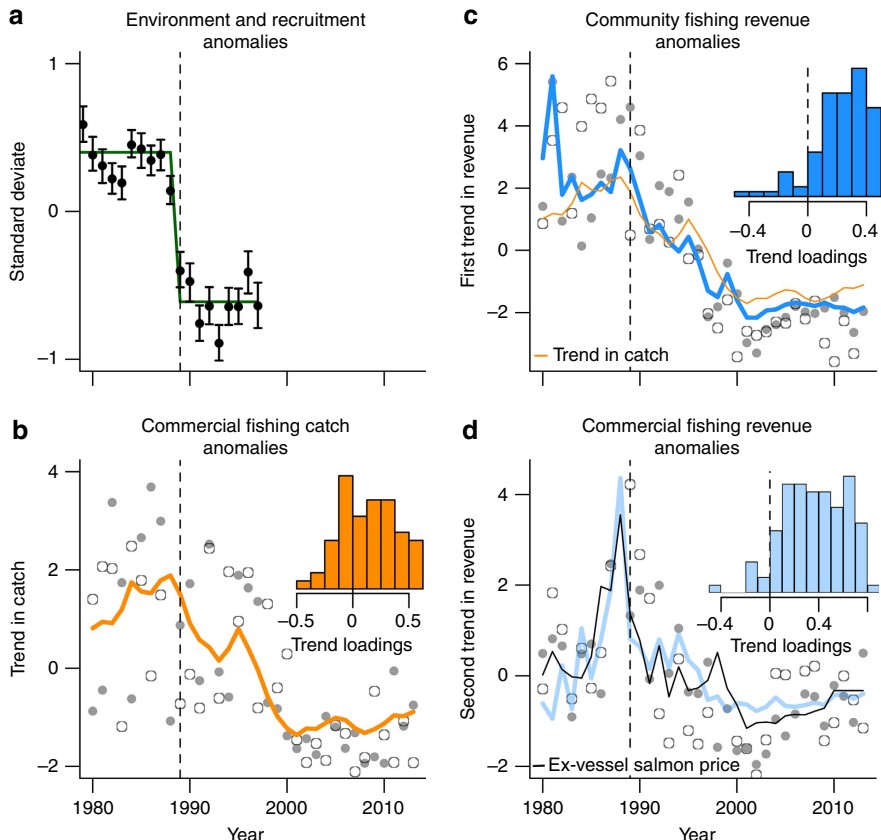

**Figure 1 | Ocean regime changes and market shifts impact fisheries catch and revenue.** (**a**) Regime shift analysis of 100 physical, chemical and biological time series from across the North Pacific region (redrawn from Hare and Mantua 2000). The green line represents the average deviate from before and after 1989; (**b**) The most commonly shared trend in commercial harvest by stock (species location combinations) (orange line) as determined by dynamic factor analysis (DFA). The analysis included time series of commercial harvest from 60 stocks delineated by fishery group (for example, crab, salmon) and location harvested (for example, Prince William Sound, Southeast). Open and closed circles in grey are example fits for two stocks. Inset histogram of factor loadings of individual stocks on the shared trend in commercial catches. Loadings indicate the strength of association between the commercial catch of a stock and the estimated trend. A negative loading indicates that the commercial catch of a stock trends in the opposite direction. (**c**) The first of two commonly shared trends in commercial fishing revenue across communities in Alaska as determined from a DFA (blue line). The analysis included time series of commercial fishing revenue from 105 communities from across Alaska. The orange line is the common trend in commercial fisheries catches by stock (**b**). Open and closed circles in grey are example fits for two communities that loaded strongly on the first trend. Inset histogram of factor loadings of individual fishing communities on the first shared trend in commercial fishing revenue. (**d**) The second of two commonly shared trends in commercial fishing revenue (blue line). Open and closed circles in grey are example fits for two communities that loaded strongly on the second trend. The black line indicates the real ex-vessel price per pound for salmon paid to fishers in Alaska. Price is z-score standardized for display on the same axes. All price and revenue data were adjusted for inflation using the consumer price index (2013 US$). Inset histogram of factor loadings of individual fishing communities on the second shared trend. The vertical dashed line in all panels marks 1989 and indicates the timing of the regime shifts.

specifically for time series, we searched for the most common shared trend among time series of commercial catches from 60 different fish stocks harvested throughout Alaska (Fig. 1b, Supplementary Table 1). The dominant pattern in commercial fisheries catch shared among stocks shows striking changes around 1989 indicating the influence of the regime shift on fishery catches (Fig. 1b). However, not all stocks or species responded in the same direction (Fig. 1b inset). While total pounds harvested in Alaskan fisheries increased by about 30% after 1989, many stocks (66%) exhibit a negative response to the changes in ocean conditions after 1989 (Fig. 1b inset). Notably, stocks of salmon and herring in the northern Bering Sea declined after 1989, but catches of walleye pollock and salmon in Southeast Alaska increased. How these shifts in the composition of commercially available species manifest as changes in revenue for resource-dependent communities is likely mediated by the exploitation strategies or constraints on individual communities.

**Ocean regime and market shifts impact fishing revenues.** To understand the impacts of the regime shift and changes in commercial fishery harvest on Alaskan fishing communities, we evaluated the trends in commercial fishing revenue which also changed substantially after the 1989 regime shift (Fig. 1c,d). We again used DFA on revenue time series for 105 Alaskan fishing communities, based on ex-vessel values, to reveal common trends in revenue that were shared among Alaska's fishing communities (Fig. 1c,d). There was model support for two independent trends in revenue over time (Supplementary Table 2), both of which indicate significant changes in revenue derived from commercial fishing, but for different reasons. The first shared trend is highly correlated ($r = 0.86$, Pearson correlation) with the common trend in commercial catches (Fig. 1b). The second shared trend is highly correlated ($r = 0.87$, Pearson correlation) with the price-per-pound paid to fishers for Pacific salmon species, the dominant commercial fisheries by value in Alaska (Fig. 1c). There are obvious feedbacks between catch and price as supply and demand

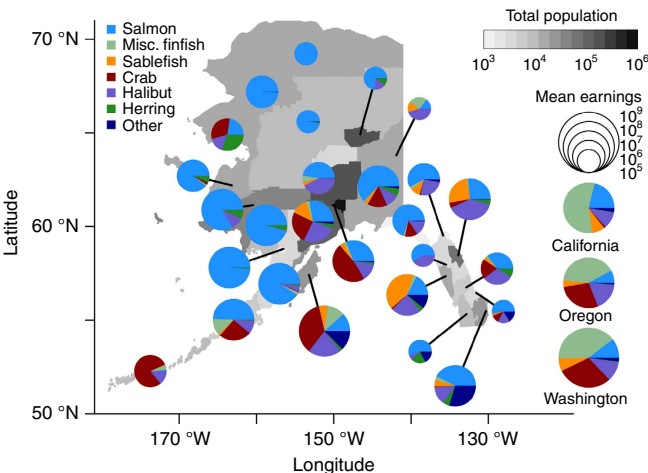

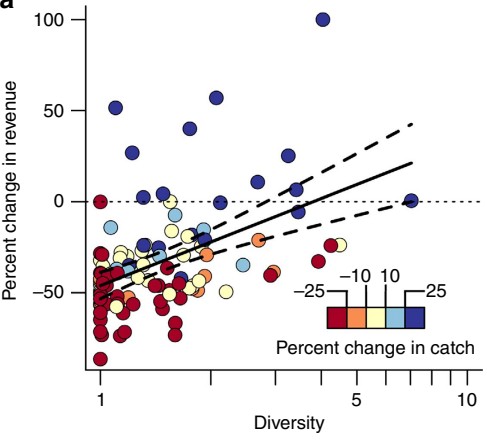

**Figure 2 | Variation in diversification of fishing opportunities across Alaskan communities.** Average revenues from the largest six major species/groups in Alaska from 1980–2013 are aggregated by census area or borough. Pie chart diameter reflects the average annual revenue (mean earnings) from fishing in 2013 US dollars. Graphs for California, Oregon and Washington are for revenue from Alaskan state fisheries by permit holders residing in those states. Revenue is allocated to permit holder residence, not fishing area.

relationships have strong influence on the value of fisheries[9]; however, this change in salmon price was driven primarily by a substantial increase in global aquaculture production of salmon[10]. While changes in catch or price alone can generate strong changes in revenue, synergistically they can influence revenue with no absolute change in either one. Simply changing the composition of catch among the different relative values can significantly impact revenue. Here we find that changes in revenue are significantly controlled by changes in catch, price and the product of the two (Supplementary Table 3).

**Diversity of fishing among Alaskan fishing communities.** A gradient in the diversity of fishing opportunities exists across Alaskan fishing communities that is driven by demographic and geographic patterns. Commercial fishing in Alaska is a vital source of income statewide providing $1.3 billion annually from harvest alone[31], and in some remote areas where other sources of income are scarce, fishing may be the only major industry[32]. Despite the clear importance of fishing and the risks associated with fishing as an income source[6], many communities participate in only a few fisheries (Fig. 2). In more developed regions, which have more access to capital or a higher diversity of commercial fish stocks nearby, communities often participate in 30 or more fisheries distributed across several species. Many communities participate in a similar group of fisheries, but few have their efforts broadly distributed across those fisheries (Fig. 2). Both the number of fisheries a community participates in and distribution among these (evenness) are potentially important dimensions of diversity and its role in stabilizing community revenues.

**Diversity and turnover provide resilience to regime changes.** To measure the effect of diversification of fishing opportunities in creating resilient communities, we compared changes in revenue in response to the regime shifts against the degree of diversification in fishing communities across Alaska. Overall, 93 of 105 communities showed decreased fishing revenue following the 1989 regime shift and abrupt change in price, but the magnitude

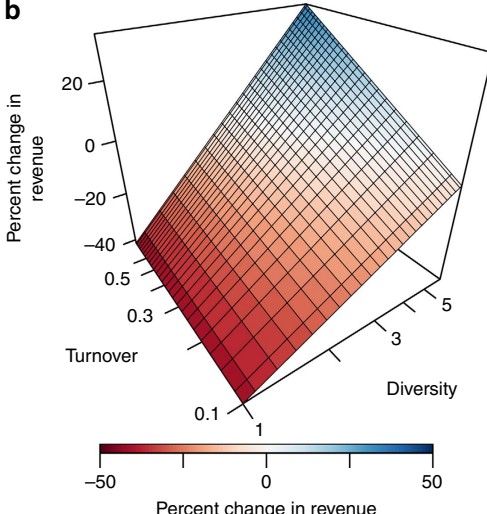

**Figure 3 | Diversification and turnover of fishing opportunities buffer against abrupt shifts.** (**a**) Observed changes in revenue from before (1980–1989) and after (1990–1999) the regime shifts are plotted against diversity of fishing opportunities for 105 Alaskan fishing communities. Diversity, plotted on a log-scale, was measured as the reciprocal Simpson's index so increasing values are increases in diversity. Diversity indicates both the total number of opportunities and the distribution among component stocks. Diversity is computed for each community over the 10 years before 1989. Total annual revenue from fishing (2013 US dollars) were used to weight years in the computation of the diversity index. The line of best fit with 95% confidence intervals is plotted in black ($R^2 = 0.21$, $P < 0.001$, linear regression). Per cent change in commercial fishing catch is represented by the colour gradient, with cool colours representing a positive per cent change and warm colours a negative per cent change. (**b**) Wireframe of the per cent change in revenue against diversity and turnover. Turnover (often referred to as beta diversity) is Jaccard's dissimilarity index measuring changes in the in the relative contribution of stocks to a communities overall portfolio (by catch). Turnover is plotted on a log scale but values range from zero turnover to 0.4 or 40% turnover, with increasing turnover towards the back of the plot. The wireframe is generated from predictions of a linear model between diversity and turnover including a significant interaction ($R^2 = 0.26$, $P < 0.0001$, linear regression, Supplementary Tables 5 and 6). Ranges used to generate the surface are well within the bounds of the data as to produce conservative changes in revenue within the bounds of the observed data (Supplementary Fig. 1).

of the response was strongly modulated by diversity in the composition of each community's revenue (Fig. 3a). We compared the average revenue in the 10 years before 1989 with

the average revenue in the 10 years following, and related the per cent change in revenue with each community's level of diversification across species and locations before the regime shift (Fig. 3a). Many communities lost >50 per cent of their revenue following the 1989 regime shifts. Others saw large increases. Communities that participated in, and had revenue spread across a broader suite of fisheries showed smaller losses in revenue in response to the ecological and market changes that occurred in 1989 (Fig. 3a). Communities with the highest levels of diversity saw little or no change in revenue. In aggregate, fishery revenues statewide declined by about 15%; therefore, a more diversified approach for many small communities could have reduced their revenue losses. An important aspect of this log-linear relationship is that a relatively small level of diversity, a value of approximately three (reciprocal Simpson's index), provides most of the benefit of diversification (Fig. 3a). Yet >80 percent of communities fall below three and most are below two. For context, this value could correspond to a strategy with three fisheries evenly divided or a dominant fishery with several smaller supporting fisheries. Here a more even distribution among fisheries provided a greater benefit than the number of fisheries alone (Supplementary Table 4). One of the major requirements for diversity to enhance stability is that components vary asynchronously. Simply diversifying among three similar species or across small spatial scales may not necessarily improve resilience.

A second consequence of this log-linear relationship is that further increases in diversity may have diminishing returns. There is a need for more explicit assessments of how to distribute risk by constructing efficient resource portfolios in resource-dependent communities. Diversification of fishing opportunities, in this coupled human-natural system, substantially reduced the vulnerability of fisheries revenues into communities against these regime shifts, demonstrating the potential for resource heterogeneity to ameliorate risk to future abrupt changes in ecosystems and markets.

A major consequence of the 1989 regime shift was complementary changes in species abundance[27], where some species increased in abundance as others showed marked declines. We measured turnover, the relative change in the composition of a community's catches, within each community's fishing portfolio from before to after 1989 using Jaccard's dissimilarity index. Together with diversification, turnover explained 26% of the variation in changes in community revenue including a significant interaction (Fig. 3b, Supplementary Tables 5 and 6). Communities with the highest diversity had higher turnover while communities with small diversity had little or no turnover. Having a diverse portfolio of fishing opportunities and the ability to shift the relative distribution of stocks within the portfolio improved changes in revenue and in fact enabled some communities to increase revenue streams from fishing despite substantial shifts in abundance, species composition, and prices (Fig. 3b). While only a small level of diversity reduces vulnerability, having a large set of fishing options provides opportunity to more easily adapt to and take advantage of fluctuations in species abundance and prices. Creating and maintaining flexibility in fishing opportunities appears to be an effective tool for providing resilience against unforeseen changes in environmental and economic conditions.

## Discussion

Diversification has long been a critical attribute of traditional fishing economies[11]. For subsistence fishing communities around the North Pacific, salmon have often been the core of their economies, but intertidal shellfish, and marine fish and marine mammals were often important, especially when salmon returns were low[33]. Recognizing the tendency of open-access fisheries to develop too many fishing boats, and in response to depleted fish stocks, 'limited entry' was introduced, which has reduced access for many people and forced specialization for efficiency. As access to fisheries is limited, buying into additional fisheries can be difficult and costly. However, the results presented here suggest that such investments likely have substantial long-term benefits to communities.

Diversification has clear benefits to fishing communities, but it can be challenging to achieve. Key limitations, especially for small communities, are access to capital to purchase fishing rights and permits, and to develop capacity for processing seafood products, and physical access to a wide array of fishing resources. Not all Alaskan stocks are fully exploited and rights for many fisheries can be purchased in markets, yet these fishing permits and rights can be prohibitively expensive. For example, permits for access to some Alaskan salmon fishing opportunities can be in excess of $300,000 (USD). In Alaska, we see patterns of diversification that follow geographic access to resources as well as economic development (Fig. 2). Communities in Southeast Alaska where diverse fishing opportunities exist in close proximity had the highest levels of diversity. Importantly, the small level of diversity needed to mitigate most of the risk in this social-ecological system (due to the log-linear relationship) could markedly reduce these barriers to diversification.

Diversification also carries risks such as loss of efficiency due to lack of specific knowledge as well as high capital costs. Diversification has been mostly thought of as an individual level strategy[6] not a community one, therefore, a community's ability to influence diversification has been limited. Yet, fishing by individuals provides a vital revenue stream into communities, which supports many other supporting industries through commerce as well as infrastructure through taxes. Importantly, the constraints on community diversification are likely to be different than those on individuals. There are often legal restrictions that may prevent individuals from owning diverse fishing rights. High capital costs, which may be prohibitive for individuals, could be reduced within communities through directed efforts such as permit pooling or lease programs. For example, Alaska Native Corporations or economic development organizations (organizations whose goals are to improve the economic standing of native Alaskans and specific regions within the state) currently use some of their resources to help people become more competitive in specific fisheries. An alternative strategy could be to use resources to help communities with limited access to capital gain more diverse fishing opportunities. Could communities circumvent some of the loss in efficiency by distributing specialist fishers among diverse opportunities, rather than having many diverse individuals? The benefits of diversification are clear and fishing dependent communities need policies and potentially novel practices that enable opportunities and encourage diversification and flexibility.

Over the last decade, there has been considerable interest and effort towards implementing ecosystem-based fisheries management under which multiple objectives and stakeholders, rather than single species, are considered in the management of fishing sectors and ecosystems[34]. Diversification lends itself nicely to an approach that calls for exploitation distributed across ecosystem components[35], while creating important buffers against variability and regime shifts. This approach may also have ecosystem benefits. As some stocks get smaller and the costs of harvest increase, switching among fishing opportunities to more abundant stocks may reduce pressure on stocks in decline.

Here we show that diversification and the ability to opportunistically shift the composition of species exploited can buffer communities against unexpected and large-scale ocean

regime shifts and changing markets. Ecosystems can undergo significant shifts in composition, but often maintain their overall productivity. While delivery of ecosystem services is provided primarily by abundant species[36], the species that are abundant now may not be in the future. Additionally, species that have small market share now may demand premium prices in the future. Diversification provides opportunities to take advantage of emerging changes and buffers against disaster; it is a tangible means to increase the resilience and adaptive capacity of coupled human-natural systems. Due to the immense complexity of the ecosystems, markets, and the feedbacks between them, we will always have limited scientific capacity to anticipate future changes. Thus, there is critical need to employ strategies that enable sustainable communities despite deep and irreducible uncertainty about the future conditions of the coupled human-natural system they depend on.

## Methods

**Data.** The state of Alaska manages all fisheries operating within three nautical miles of shore and a small set of fisheries operating in the USA exclusive economic zones (>3 nautical miles). Alaskan commercial fisheries are delineated by taxa (either species or species group), the area harvested, and the gear used to harvest them. State managed fisheries are mostly managed using some form of limited entry where access is controlled, but can be bought or sold among parties. Any individual harvesting in a state managed fishery requires a permit specific to the taxa, area fished and gear used. All landed catches and revenue in Alaskan commercial fisheries are required to be reported. The Commercial Fisheries Entry Commission (CFEC) tracks landings and revenue by permit and permit-holder residency. Therefore, reported catch and revenue are assigned to communities where the permit holder resides. For confidentiality reasons, the CFEC cannot report very small fisheries that have less than four permit holders per community. Therefore, these fisheries are not included in our analysis. Management of sablefish and halibut fisheries switched to individual transferable quota systems in 1995, which could alter the revenue and catches in these fisheries. However, excluding catches and revenues for these species does not change the results in this study.

We used community level gross fishing revenues data collated by the CFEC. These revenues represent the sum of annual gross fishing revenues attributable to CFEC commercial fishing permits registered to a specific Alaskan community. Some communities have seen changes and permit migration. We ran these analyses using revenue per permit fished to account for these changes, and there is no change in the results or conclusions. All revenue data were adjusted for inflation to 2013 USD using the US annual average consumer price index (http://www.bls.gov/cpi/). To evaluate spatial patterns in diversification, we used revenue data aggregated to USA census areas (Fig. 2). Population and demographic information by census area were taken from USA 2010 census data (www.census.gov). Salmon price data was provided by CFEC summary data. Prices were adjusted for inflation to 2013 USD using the USA annual average consumer price index.

**Analysis.** To search for common trends in commercial fisheries catch among different stocks across Alaska as well as common patterns in community revenues we applied DFA[30] to time series of commercial harvest by stock and to time series of Alaskan fishing community revenue (separate analyses). Similar to principal component analysis, DFA is a dimension reduction technique designed specifically for use with time series data. With DFA, we are trying to explain temporal variation in a set of $n$ observed time series using linear combinations of independent hidden random walks. The model structure is as follows:

The state equation of a vector of common random walk trends over time:

$$\boldsymbol{x}_t = \boldsymbol{x}_{t-1} + \boldsymbol{w}_t \qquad \boldsymbol{w}_t \sim MVN(0, \boldsymbol{Q}) \qquad (1)$$

Observation equation relates trend ($\boldsymbol{x}$) to observations ($\boldsymbol{y}$).

$$\boldsymbol{y}_t = \boldsymbol{Z}\boldsymbol{x}_t + \boldsymbol{v}_t \qquad \boldsymbol{v}_t \sim MVN(0, \boldsymbol{R}) \qquad (2)$$

Here the vector of observations for catches in each stock (or revenue in each community) at time $t$ ($\boldsymbol{y}_t$) are modelled as linear combinations of hidden trend ($\boldsymbol{x}_t$) and factor loadings on the hidden trend for each stock (community) ($\boldsymbol{Z}$). $\boldsymbol{v}_t$ and $\boldsymbol{w}_t$ represent the observation and process error structures, respectively. To make the model estimable[24] process error ($\boldsymbol{Q}$) was set to a diagonal matrix of value 1. Observation errors ($\boldsymbol{R}$) are from a multivariate normal distribution. The number of trends and the error structure are determined through model selection. Candidate models were compared using AIC based on the maximum likelihood of the model fit. Model selection results for DFA analyses are in Supplementary Tables 1 and 2.

To evaluate common trends in commercial fisheries harvest among stocks around Alaska, we applied the DFA to time series of fisheries delineated by taxa and location harvest (that is, catch was aggregated among different gear types within the same taxa/location combination). This yielded an increased number of complete time series of catch. We limited stocks to only those that had no more than three zero-catch values ($n = 60$). This increased the overall diversity of taxa

and locations represented in the data. We compared models with one or two hidden trends as well as two error structures; where all stocks share a single observation error (diagonal and equal) and where all stocks have individual observation error (diagonal and unequal). All data were z-scored to account for differences in means and intrinsic variance dynamics.

In a separate analysis, we applied the DFA to time series of fishing community revenues allocated to Alaskan cities. For fishing communities we included only those that had non-zero revenue in all years between 1980 and 1999 ($n = 105$). We compared models with 1 or 2 hidden trends as well as two error structures; where all stocks share a single observation error (diagonal and equal) and where all stocks have individual observation error (diagonal and unequal). All data were z-scored to account for differences in means and intrinsic variance dynamics. DFA models were fit using the 'MARSS' package in R[37].

To measure the effect of diversification on changes in community revenues and commercial fishing catches we compared revenue and catch before and after 1989 and diversity in fishing opportunities as well as turnover. We included only communities with non-zero revenue in all years between 1980–1999. Shifts in revenue were simply computed as the per cent change in the mean annual revenue for the period 1980–1989 and the period 1990–1999 allocated to individual communities. Changes in catches were computed this same way using total annual pounds landed allocated to individual communities. The earliest available year for catch and revenue data reported to individual communities is 1980 (total of 10 years pre-regime shift). For an even comparison we used only the 10 years post-1989 (1990–1999).

Diversity was computed for each community over the 10 years before the 1989 (1980–1989). We used the reciprocal Simpson's index (1/D) where diversity (D)

$$D = \sum_i p_i^2 \qquad (3)$$

where $p_i$ is the proportion of total revenue by fisheries (taxa and location). To compute an overall diversity metric for each community we weighted each year by the relative proportion of that year to the total revenue for all years.

To measure turnover in the relative composition of a community's commercial fishing catches, we compared the commercial fisheries catches from before and after 1989. For each community a relative composition by fishery was computed based catches on for each of two periods, 1980–1989 and 1990–1999. Specifically, each fishery's contribution to the total catch for each period in total was computed. Then, the relative change in composition was computed using Jaccard's dissimilarity (J) index between these two time periods, A (1980–1989) and B (1990–1999).

$$J = \frac{|A \cup B| - |A \cap B|}{|A \cup B|} \qquad (4)$$

All diversity and turnover metrics were computed in R using the 'vegan' package[38].

We compared relationships between changes in revenue and diversity and turnover using linear regression. To assess the relative strength of the two types of diversity and their interaction we computed effect sizes using z-score standardized covariates.

**Data availability statement.** The fisheries data that support the findings of this study are publically available online from the Commercial Fisheries Entry Commission (CFEC)[39].

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

## Acknowledgements

Funding for this project was provided through a NSF Dynamics of Coupled Natural and Human Systems grant (CNH #1114918), as well as a NSF Graduate Research Fellowship to T.J.C. We thank Tim Walsworth and Jake Allgeier for helpful comments on an earlier draft of this manuscript, and the Alaska CFEC for access to the data used in this analysis.

## Author contributions

T.J.C., D.E.S. and R.H. designed the study. T.J.C. analysed the data. T.J.C. wrote the paper with contributions from D.E.S. and R.H.

## Additional information

**Competing financial interests:** The authors declare no competing financial interests.

