## [Peer Review File · Nature Communications]

Reviewers' Comments:

Reviewer #1 (Remarks to the Author):

The authors examined the role of diversification and turnover in how Alaskan communities buffered a substantial ecological regime shift. In general, I thought it was a well-written paper. It is conceptually a bit similar to Aguilera et al. (2015), but has the distinct advantage of using multiple sites over time. I think the authors should explicitly emphasize this contribution (i.e. it is both spatially and temporally comparative- in short, the data we all wish we had).

It is important for the authors to note that during the study period, the governance of some Alaskan fisheries also changed profoundly, with the introduction of ITQs. This could obviously influence the diversity of opportunities fishers have because it restricts entry. I think you need to convincingly discount the potential effects of that governance change on revenue (theory would have it that the governance change would have had profound implications for revenue).

So as a social scientist, I found the article pretty lacking in engagement with the social science literature into the causes and consequences of diversification. Development scholars, geographers, economists, etc. have been studying diversification in social structures and processes (like livelihood diversification) for decades. Many social scientists view diversification as a double edge blade- it can provide buffers against shocks (as demonstrated in this article), but also can be a trap. We often see increasing specialisation at the individual/household level along a development gradient, but that diversity remains at the community level. I suppose I'd like to see a more grounded discussion of diversification in fishing communities based on some of the foundational work. People like F. Ellis and even Eddie Allison (who is at UW!!!!). I feel like this would make the article more about the core issues, rather than about Alaska. I'd also like to see a more balanced discussion of the consequences of diversification- I am unfamiliar with the Alaskan context, but could diversification allow for opportunistic exploitation of fisheries, as per the Unsworth and Branch article a few years back?

The data and analysis are a bit tangential to my core expertise, but I found the methods clear and easy to follow. I'm unfamiliar with dynamic factor analysis, I've used factor analysis a bunch and it seems like the authors have used a perfectly appropriate tool for the problem at hand.

Fig. 1. I struggled with "Open and closed circles in gray are example fits for two stocks". Were these randomly selected or the ones that fit the trend line the best? I've normally seen panel b to

the right of panel a, not below. This threw me off. Note that in c the orange line is not dashed as indicated in the legend. How come you do not include the revenue trend in panel b? Is the "commonly shared trend" (y-axis) equivalent to a factor or principle component?

Fig 2. Might be worth mentioning in the legend that these are reported based on where the permit holder resides, not necessarily where they fish. Thus, inland communities don't actually catch halibut in the mountains.

Fig. 3b is a bit hard to get my head around, primarily because of the negative log turnover (where -0.4 at the back of the graph indicates more turnover than -0.8 at the front of the graph). What is there is correct, but confusing. Is it possible to plot the non log values there (even if on a log scale) or turn that axis around? Hope that makes sense.

Aguilera SE, Cole J, Finkbeiner EM, Le Cornu E, Ban NC, Carr MH, et al. (2015) Managing Small-Scale Commercial Fisheries for Adaptive Capacity: Insights from Dynamic Social-Ecological Drivers of Change in Monterey Bay. PLoS ONE 10(3): e0118992.
doi:10.1371/journal.pone.0118992

Reviewer #2 (Remarks to the Author):

This paper attempts to assess if diversity and turnover (which is a strange choice of term given its very different meaning in the economics literature) buffer communities from abrupt changes in resource flows and prices. It is an important and timely topic and they do somewhat fulfil this goal, however, I have several misgivings about a) the relevance of the unit of analysis, b) the lack of any theoretical grounding and motivation for the variables used, c) the assumptions made regarding measures of diversity and the lack of discussion of the multiple ways in which diversity could be conceptualized, and how it links to community resilience, and d) the interpretation of results. I elaborate on my concerns relating to each of these topics below.

a) A key problem with the paper, as I see it, is that the motivation for why communities is a relevant unit of analysis is never provided. Is there some sort of community sharing system in these communities? Else, why is community a relevant unit of analysis? Even though one community has a high diversity of fisheries, if the individual fishers within them are not diversified it is questionable how resilient the community is. E.g. in Maine it is hard for fishers to diversify because the licensing system is not flexible, and limited entry and other efforts to reduce pressure creates difficulties for fishers to get into new fisheries, creating path dependency. Reading on similar issues appear to be relevant in Alaska. While I am not dismissing the idea I find no justification for this choice in the text and without some more explanation I am not convinced the community unit used here is particularly relevant in a real

world scenario, and certainly not likely to apply beyond communities of traditional culture and sharing systems (which is most of the world).

b) and c) are actually linked and I will try to explain why. The lack of theoretical grounding and motivation for the variables used is problematic. It makes the paper feel somewhat shallow and naïve. At the onset diversity and turnover are claimed to be of key importance for stabilizing community economies against shocks over time. However, turnover is not even defined until half way through the paper and when it is no explanation of how the concept is linked (conceptually/theoretically) to the dependent variable is provided. One has to glean it from between the lines. Diversity is a concept used widely and as such has many different meanings. In the context of fisheries it could be the type of spp, geographic location and habitat type, seasonality, but the authors seem to settle on just counting diversity as numbers of fisheries communities are involved in. I find this problematic for two reasons. First, simply because it neglects to engage the many different possible diversities that could apply and be relevant, but second and more important, because the authors never provide any theoretical or conceptual motivation for how (i.e. a theory of causality) and why diversity should be linked to community resilience. Without this it even makes it hard to evaluate if the authors operationalization of the variable 'diversity' is relevant, and thus if the results and conclusions are valid.

d) Some of the interpretation of results is actually misleading in my view. The most striking example is the discussion of the benefits associated with involvement in diverse fisheries. On line 151-52 authors say that "communities with the highest levels of diversity saw little or no change in revenue", suggesting a u-shaped response and the fact that beyond a certain point being involved in more fisheries does not pay off. This seems like a fairly important finding but it is not brought up again, and results are discussed as if the relationship between diversity and revenue was linear and making statements like "The benefits of diversity are clear and fishing dependent communities need policy that enables opportunities and encourages diversity and flexibility in fishing communities". The fact that there is not a linear relationship between diversity and income suggest that beyond a certain point resources should maybe be put towards other things than diversification, but this is not the message promoted by the authors.

Furthermore, I still feel it is an overly simplistic treatment of diversity and its possible contribution to sustained income of communities. E.g. on line 154-157 authors say that the max revenues accrue to communities involved in two fisheries with 50/50 distribution of ...effort or volume (the latter is not clear). But is it really about just any two fisheries, or is the key not in which two types of fisheries? Two ground fish fisheries or two pelagic or two benthic shellfish fisheries are unlikely to provide the same robustness to shock as being involved across these sectors, no? but this is never even discussed let alone analyzed.

Also, is it possible that the reason why some communities who were involved in multiple fisheries (which not many others had access to) sustained their revenue was exactly because not everyone had access to them? If everyone had been diversified, the benefit derived to any community from the additional fisheries would certainly have been much less and the risk of just

shifting exploitation pressure to the newly emerging resource would be high.

A more general note is that I find the text unfulfilling in that repeatedly issues of potentially key importance are flagged but never followed up with a discussion (see e.g. issues related to how to conceptualize and measure diversity, line 137-140, the apparently naiveté to overcoming the limited entry system and creating flexible fisheries access systems, line 174-75).

Many more comments on specific sections of the text can be found directly in the attached document. I apologize for any spelling mistakes as the review was done on an iPad on a train journey.

Reviewer #3 (Remarks to the Author):

This paper uses catch and revenue from commercial fishing in Alaskan fishing communities to test whether diversity and turnover in the composition of fishing opportunities reduced economic instability due to regime shifts in ocean productivity and market dynamics.

Comments:

It is generally known that diversity helps to reduce uncertainty and risk. A good example is the financial sector where no portfolio manager worth his or her salt would construct a portfolio without taking into account diversity. Hence, the question posed by the authors is not intriguing. What would have been really interesting is to explore the optimal level of diversity for a common pool resource such as fishery resources.

Using catch and revenues to test whether diversity in the composition of fishing opportunities reduces economic instability is incorrect. Catch and revenues alone may result in misleading conclusions since factors such as changes in fishing cost, effort, subsidies and consumer demand are not incorporated in the analysis. In order to evaluate economic instability, a typical methodology should begin with the derivation (or assumption) of a social welfare function, which can then be used to assess measures of economic efficiency and stability. Without a bio-economic model of fishery behavior, the current analysis is essentially "non-economic" and therefore the conclusions of the paper are almost surely incorrect.

I am concerned that the analysis is based on the occurrence of just one environmental anomaly in 1989. How is one to tell that the changes we see in catch and revenues are due to this particular environmental anomaly? Were there other environmental anomalies before and after 1989? If yes how did they affect catch and revenues? A quick review of the literature indicates that other environmental anomalies occurred in the Bering Sea, for example.

Reviewers' comments:

Reviewer #1 (Remarks to the Author):

The authors examined the role of diversification and turnover in how Alaskan communities buffered a substantial ecological regime shift. In general, I thought it was a well-written paper. It is conceptually a bit similar to Aguilera et al. (2015), but has the distinct advantage of using multiple sites over time. I think the authors should explicitly emphasize this contribution (i.e. it is both spatially and temporally comparative- in short, the data we all wish we had).

Using a small scale set of fisheries and permit holders, Aguilera et al. (2015, PLoS ONE) show that adaptive capacity is important for buffering fisheries against swings in productivity. This is an interesting analysis that we were not previously aware of (we now cite this paper in the introduction). Our dataset uses wide spread regime shifts in fish productivity and natural resource markets that affect more than 60 fisheries, more than 100 communities, and thousands of permit holders spread across Alaska. We now emphasize the distinctiveness of our data set for addressing the role of diversification and turnover of livelihoods for resilience in social-ecological systems (Lines 83-84 and 98-100).

It is important for the authors to note that during the study period, the governance of some Alaskan fisheries also changed profoundly, with the introduction of ITQs. This could obviously influence the diversity of opportunities fishers have because it restricts entry. I think you need to convincingly discount the potential effects of that governance change on revenue (theory would have it that the governance change would have had profound implications for revenue).

The introduction of ITQ's for Alaskan halibut and sablefish occurred in 1995. We re-analyzed the data by excluding these two species entirely from the catch data analyses as well as the revenue for communities. This does not affect the results or conclusions of these analyses. Therefore, the responses we report are not generated from a change in management. We now include this in the description of the data in the methods section (Lines 421-424).

So as a social scientist, I found the article pretty lacking in engagement with the social science literature into the causes and consequences of diversification. Development scholars, geographers, economists, etc. have been studying diversification in social structures and processes (like livelihood diversification) for decades. Many social scientists view diversification as a double edge blade- it can provide buffers against shocks (as demonstrated in this article), but also can be a trap. We often see increasing specialization at the individual/household level along a development gradient, but that diversity remains at the community level. I suppose I'd like to see a more grounded discussion of diversification in fishing communities based on some of the foundational work. People like F. Ellis and even Eddie Allison (who is at UW!!!!). I feel like this would make the article more about the core issues, rather than about Alaska. I'd also like

to see a more balanced discussion of the consequences of diversification- I am unfamiliar with the Alaskan context, but could diversification allow for opportunistic exploitation of fisheries, as per the Unsworth and Branch article a few years back?

We expanded our discussion of the potential consequences of diversification as well at the opportunities that might exist in community level diversity that might not be available to individuals (Lines 229-256).

The reviewer brings up the idea of opportunistic exploitation as a pathway to overexploitation (Branch et al. 2013, Trends in Ecology and Evolution). The basic idea is that rare species could be harvested opportunistically while fishers are primarily targeting a more abundant species. Thus while rare species fished on their own may not be profitable, opportunistically harvested they can be very valuable. This would not lead to overexploitation in our system as all commercially harvested species are regulated through some sort of harvest limits. Fishers would need permits for all species harvested and would be limited within each species' allowable catch, as set by Alaska Department of Fish and Game.

The data and analysis are a bit tangential to my core expertise, but I found the methods clear and easy to follow. I'm unfamiliar with dynamic factor analysis, I've used factor analysis a bunch and it seems like the authors have used a perfectly appropriate tool for the problem at hand.

We agree with this statement. DFA was developed in the economics literature to provide a tool for finding common patterns and quantifying interactions in an explicit time-series framework. It is only beginning to be used in more ecosystem analyses but its utility in this regard has become clear. We believe it is a powerful way to demonstrate the changes in ecosystem structure that occurred with the regime shift we are exploring here.

Fig. 1. I struggled with "Open and closed circles in gray are example fits for two stocks". Were these randomly selected or the ones that fit the trend line the best?

We now clarify that the selected example data sets are ones that had strong loadings on the common trend. Many time series have little or no loadings on the trend and therefore are not well represented by the common trend.

I've normally seen panel b to the right of panel a, not below. This threw me off.

The order of the panels is important for the flow of the ideas, and we want the two revenue trends to be one on top of the other for easy interpretation of the timing of the events.

Note that in c the orange line is not dashed as indicated in the legend.

We have fixed the caption to correctly indicate the line types.

How come you do not include the revenue trend in panel b?

Our intent here is to show that an environmental regime shift affects catches, and that catches affect revenue. However, there is no strong evidence that revenue affects catches. For example, most salmon fisheries lost substantial revenue due to a regime shift in the prices paid to fishers after 1989. Yet for most salmon fisheries there is little or no change in catch.

Is the "commonly shared trend" (y-axis) equivalent to a factor or principle component?

We now state that the commonly shared trend is analogous to a principal component, but is cast formally in a time-series context, in the Fig. 1 caption.

Fig 2. Might be worth mentioning in the legend that these are reported based on where the permit holder resides, not necessarily where they fish. Thus, inland communities don't actually catch halibut in the mountains.

We now state in the legend that revenue is allocated to permit holder residence, not by fishing area (Figure 2 caption)

Fig. 3b is a bit hard to get my head around, primarily because of the negative log turnover (where -0.4 at the back of the graph indicates more turnover than -0.8 at the front of the graph). What is there is correct, but confusing. Is it possible to plot the non log values there (even if on a log scale) or turn that axis around? Hope that makes sense.

We have restructured and adjusted the labels on the axis to be non-log values to ease interpretation.

Reviewer #2 (Remarks to the Author):

This paper attempts to assess if diversity and turnover (which is a strange choice of term given its very different meaning in the economics literature) buffer communities from abrupt changes in resource flows and prices.

In the investment theory, 'turnover' refers to the fraction of an investment portfolio that is sold in a particular month or year (Wermers 2000, Journal of Finance). Our use of diversification and turnover of fishing opportunities are analogous to investment portfolios and we have tried to draw this parallel as clearly as possible in this paper.

It is an important and timely topic and they do somewhat fulfill this goal, however, I have several misgivings about a) the relevance of the unit of analysis, b) the lack of any theoretical grounding and motivation for the variables used, c) the assumptions made regarding measures of diversity and the lack of discussion of the multiple ways in which diversity could be conceptualized, and how it links to community resilience, and d) the interpretation of results. I elaborate on my concerns relating to each of these topics below.

a) A key problem with the paper, as I see it, is that the motivation for why communities is a relevant unit of analysis is never provided. Is there some sort of community sharing system in these communities? Else, why is community a relevant unit of analysis? Even though one community has a high diversity of fisheries, if the individual fishers within them are not diversified it is questionable how resilient the community is. E.g. in Maine it is hard for fishers to diversify because the licensing system is not flexible, and limited entry and other efforts to reduce pressure creates difficulties for fishers to get into new fisheries, creating path dependency. Reading on similar issues appear to be relevant in Alaska. While I am not dismissing the idea I find no justification for this choice in the text and without some more explanation I am not convinced the community unit used here is particularly relevant in a real world scenario, and certainly not likely to apply beyond communities of traditional culture and sharing systems (which is most of the world).

In these resource dependent communities, reliable returns produced from fishing support other economic activities and employment both directly and indirectly related to the resource extraction such as processing, boat building and maintenance, fuel provisioning and other basic necessities. We now include this justification for the community as a relevant unit to study these impacts in the text (Lines 232-235). Also, many community level revenue streams (e.g., revenues for schools) are linked directly to taxes on fishery production.

Additionally, the reviewer is correct that diversification is typically an individual action. In our discussion of the challenges for diversification, we acknowledge this challenge (Lines 229-239). The importance of community level diversification is a major result of this analysis. We now emphasize that in the discussion (Lines 229-239). Licensing restrictions can prevent individual diversification, and analogous issues are found in Alaskan fisheries management. This lends even more importance to the concept of community diversification, whereby collectively communities could circumvent individual licensing restrictions and buffer their livelihoods against instability and change in the resources and the markets.

b) and c) are actually linked and I will try to explain why. The lack of theoretical grounding and motivation for the variables used is problematic. It makes the paper feel somewhat shallow and naïve. At the onset diversity and turnover are claimed to be of key importance for stabilizing community economies against shocks over time. However, turnover is not even defined until half way through the paper and when it is no explanation of how the concept is linked (conceptually/theoretically) to the dependent

variable is provided. one has to glean it from between the lines. Diversity is a concept used widely and as such has many different meanings. In the context of fisheries it could be the type of spp, geographic location and habitat type, seasonality, but the authors seem to settle on just counting diversity as numbers of fisheries communities are involved in. I find this problematic for two reasons. First, simply because it neglects to engage the many different possible diversities that could apply and be relevant, but second and more important, because the authors never provide any theoretical or conceptual motivation for how (i.e. a theory of causality) and why diversity should be linked to community resilience. Without this is even makes it hard to evaluate if the authors operationalization of the variable 'diversity' is relevant, and thus if the results and conclusions are valid.

The reviewer suggests there is a lack of introduction to the types of diversity and what is meant by turnover. We agree that we did not adequately present the methods we evaluated. We now introduce the idea of turnover early in the paper (Lines 27, 55-56, 64-65) and make direct links to its use in finance. We also moved more explanation of our evaluation of diversification, both in terms of richness or the number of fisheries and diversity (which includes the evenness of spread across fisheries) from the methods to the beginning of the paper (Lines 79-81). We summarize those changes here.

We evaluated the diversity of fishing opportunities in two ways: the number of fisheries they participate in and using an index measure (Simpson's index) that incorporates the number of fisheries and the distribution among fisheries as well (Lines 151-153). We found that diversity accounting for the distribution among fisheries provided more benefit than simply the number of fisheries a community participated in. This is a key finding of our analysis. We now include a simple regression table comparing the explanatory power of these two types of diversity in the supplement (Table S5) and discuss these results on lines 165-169

Please note that we have broken the following long comment into several pieces to emphasize how we respond to these concerns.

d) Some of the interpretation of results is actually misleading in my view. The most striking example is the discussion of the benefits associated with involvement in diverse fisheries. On line 151-52 authors say that "communities with the highest levels of diversity saw little or no change in revenue", suggesting a u-shaped response and the fact that beyond a certain point being involved in more fisheries does not pay off.

The reviewer is correct that we emphasize a saturating relationship (we are assuming this is what they mean by 'U-shaped') between revenue change and diversity (Lines 163-164). The form of this relationship tells us two important things. First, only a small level of diversity is required to realize most of the benefit (Lines 163-166). This is critical in our system as more than 80% of communities fall below a diversity value of 2. Second, as the reviewer states, there may be diminishing returns from continuing to increase diversity. However, in our system, high levels of diversity provide opportunities to easily adapt to fluctuations (Figure 3, Lines 180-182). Therefore, it is not clear that

there is less value in high diversity. We have improved our discussion around these points (Lines 159-191).

This seems like a fairly important finding but it is not brought up again, and results are discussed as if the relationship between diversity and revenue was linear and making statements like "The benefits of diversity are clear and fishing dependent communities need policy that enables opportunities and encourages diversity and flexibility in fishing communities". The fact that there is not a linear relationship between diversity and income suggest that beyond a certain point resources should maybe be put towards other things than diversification, but this is not the message promoted by the authors.

This is a very important point that we were trying to communicate in our paper – but was obviously not clear. We have edited lines 159-191 in an attempt to make this point clearer. The point being that increases in diversity are most beneficial to communities that currently have little diversity in the fisheries they rely upon. As diversity increases, there becomes a point of diminishing returns.

Furthermore, I still feel it is an overly simplistic treatment of diversity and its possible contribution to sustained income of communities. E.g. on line 154-157 authors say that the max revenues accrue to communities involved in two fisheries with 50/50 distribution of ...effort or volume (the latter is not clear). But is it really about just any two fisheries, or is the key not in which two types of fisheries? Two ground fish fisheries or two pelagic or two benthic shellfish fisheries are unlikely to provide the same robustness to shock as being involved across these sectors, no? but this is never even discussed let alone analyzed.

We provided this example as a context for evaluating how a community might achieve a particular diversity index value. We now discuss how asymmetric dynamics among fishing opportunities is a requirement for diversity to improve stability, and that this may not be achieved with diversity only in similar species or small spatial scales (Lines 163-168). The goal of this study is not to design optimal portfolios but rather, to highlight the magnitude of the observed relationship between diversity and resilience. We also added statement for the need for more explicit assessments about how to distribute risk in resource-dependent communities (Line 167-168).

Also, is it possible that the reason why some communities who were involved in multiple fisheries (which not many others had access to) sustained their revenue was exactly because not everyone had access to them? If everyone had been diversified, the benefit derived to any community from the additional fisheries would certainly have been much less and the risk of just shifting exploitation pressure to the newly emerging resource would be high.

We re-analyzed the data by aggregating revenues to the entire state, which showed that Alaska-wide revenues declined only 15% in response to regime shifts. A more diversified approach for many small communities could have reduced their revenue

losses from more than 50% loss to only 25% loss or better. Not all Alaskan stocks are fully exploited therefore opportunities exist for increasing diversity that are not at the expense of other communities. We have added these points to the manuscript on lines 157-158 and 203-204

A more general note is that I find the text unfulfilling in that repeatedly issues of potentially key importance are flagged but never followed up with a discussion (see e.g. issues related to how to conceptualize and measure diversity, line 137-140, the apparently naïveté to overcoming the limited entry system and creating flexible fisheries access systems, line 174-75).

We have expanded our explanation of the different measures of diversity as well as the results of their relative role in increasing community resilience (Lines 165-171).

Reviewer #3 (Remarks to the Author):

This paper uses catch and revenue from commercial fishing in Alaskan fishing communities to test whether diversity and turnover in the composition of fishing opportunities reduced economic instability due to regime shifts in ocean productivity and market dynamics.

Comments:

It is generally known that diversity helps to reduce uncertainty and risk. A good example is the financial sector where no portfolio manager worth his or her salt would construct a portfolio without taking into account diversity. Hence, the question posed by the authors is not intriguing. What would have been really interesting is to explore the optimal level of diversity for a common pool resource such as fishery resources.

While the value of diversification for reducing risks is clear in financial markets, its value to communities dependent on naturally-fluctuating ecosystem resources is less appreciated especially when considered from a management perspective. Also, there are few empirical analyses showing how diversity can increase community resilience, especially at the scale used in this study. It is important to also note that diversification does not reduce uncertainties, it changes the risk of those uncertainties. This is why diversification is such a common and effective strategy in investment, and we show here that it also has analogous effects on resource flows from marine ecosystems to people.

A thorough analysis of optimum diversity for constructing fishing portfolios would almost certainly be best performed in a theoretical modeling framework, which is outside the scope of this study. In this study, an empirical analysis, we do discuss the level of diversification that produces a significant benefit to communities (Lines 163-166). It is our hope that evidence such as this will inspire new research about how to

design and manage resource exploitation strategies for communities as they attempt to cope with the enormous uncertainties associated with ongoing global and local change.

Using catch and revenues to test whether diversity in the composition of fishing opportunities reduces economic instability is incorrect. Catch and revenues alone may result in misleading conclusions since factors such as changes in fishing cost, effort, subsidies and consumer demand are not incorporated in the analysis. In order to evaluate economic instability, a typical methodology should begin with the derivation (or assumption) of a social welfare function, which can then be used to assess measures of economic efficiency and stability. Without a bio-economic model of fishery behavior, the current analysis is essentially "non-economic" and therefore the conclusions of the paper are almost surely incorrect.

The point of this comment is really not clear. There are many ways to approach a scientific problem. The way we have approached this problem is to evaluate changes in revenue as an important aspect of the economic stability. There is significant value for fishers and communities of stable revenue streams. Additionally, the data on costs do not exist at the scale of these analyses and we have accounted for changes in effort by assessing the community revenues per permit fished. This does not change the results or conclusions (Lines 427-429).

I am concerned that the analysis is based on the occurrence of just one environmental anomaly in 1989. How is one to tell that the changes we see in catch and revenues are due to this particular environmental anomaly? Were there other environmental anomalies before and after 1989? If yes how did they affect catch and revenues? A quick review of the literature indicates that other environmental anomalies occurred in the Bering Sea, for example.

It is evident that ecosystems show temporal variability across a range of time scales; from seasonal, to inter-annual, to large scale abrupt shifts that persist for decades (so-called regime shifts). Variability in the environment and species abundances are a challenge to all natural resource exploitation and management. But, regime shifts, like the changes in 1989 are not small environmental anomalies - they are large in scale, both in the space (affecting most of the North Pacific) and time (changes are persistent for decades) that involve fundamental reorganization in the composition and productivity of species that characterize the ecosystem. These persistent changes seen across many environmental indices (Figure 1 A) are reflected in changes in catch and revenue for Alaskan fishing communities (Figure 1 B,C)

An enormous and influential literature has developed in the last two decades that describes the tendency of regime shifts to characterize the dynamics of ecosystems, thereby making communities dependent on them prone to collapse (e.g. Scheffer et al. 2001, Nature, cited over 4000 times). The literature that has developed about how people should deal with impending regime shifts is to focus on early warning signals (e.g. Scheffer et al. 2009, Nature). However, others have posited that increased heterogeneity in society and in ecosystems are one way to develop resilience to regime

shifts that are difficult to predict or to respond to in prescriptive ways (e.g. Berkes et al. 2003 and the Millennium Ecosystem Assessment). We believe that our analyses here provide the first large-scale empirical demonstration of the benefits of ecosystem heterogeneity (diversity) for developing resilience to large scale regime shifts.

Reviewers' Comments:

Reviewer #2 (Remarks to the Author):

In general I Think the manuscript has improved and I appreciate that the authors have redone some of the analysis as a result of my comments.

While I note that the authors have now incorporated some more discussion on several of the topics I suggested in relation to diversification (which I Think makes for a more nuanced and interesting discussion) I felt that the results/discussion section could do with a more clear structure. The information is basically all there but I found it hard to follow at times.

Is it possible to structure in such a way that you bascially take each of the things you have looked at and lay out the pattern , what it means and why it is interesting, in one paragraph per topic - thus unpacking the problem one step at the time. Right now there isa bit of a mishmash of literature Review and results and not Always clear what was found in thsi study.

regarding their respnse to my comments:

Second respons relating to Community as a unit of analysis: This is still my biggest perceived weakness with the paper. Ok I buy the unit but still do not Think it is sufficently elaborated/theoreticáally justified in the text. What I mean is that I am looking for a deeper engagement with what Community resilience is about. I understand that tax Revenue can be important, but as a devil's advocate I could argue that having 90% offisher unemployed and 10% catching all the fish would be ok from that perspective, as long as they generate the same tax flow to the Community. but is this really true?

Also, if by including a justification you mean the lines that appears in the last paragraph of the paper (line 235 as in the response) then I do not Think that is sufficient. it should come as you explain the justification of variables and design of the study.

Authors could do a better job of linking their choice of Community diversity to resilience - some suggested readigs are:

Chapter on diversity in the book Principles of Resilience that came out a few years ago, also "Management forcing increased specialization in a fishery system" J Hentati-Sundberg, J Hjelm, WJ Boonstra, H Österblom

Ecosystems 18 (1), 45-61 - which gets into the interplay between individual diversity/specialization and system diversity.

You do not define turnover the first time it is used (line 65) which is still somehwt confusing - suggest moving the explanation up further.

Line 208-210- I dont understand what is meant?

Reviewer #3 (Remarks to the Author):

The authors have adequately addressed my comments.

I have no further comments.

Below we have included a point-by-point discussion of the changes made in response to the additional comments of the reviewers.

Reviewers' comments (in standard font, our responses in bold):

Reviewer #2 (Remarks to the Author):

In general I think the manuscript has improved and I appreciate that the authors have redone some of the analysis as a result of my comments. While I note that the authors have now incorporated some more discussion on several of the topics I suggested in relation to diversification (which I think makes for a more nuanced and interesting discussion) I felt that the results/discussion section could do with a more clear structure. The information is basically all there but I found it hard to follow at times. Is it possible to structure in such a way that you basically take each of the things you have looked at and lay out the pattern, what it means and why it is interesting, in one paragraph per topic - thus unpacking the problem one step at the time. Right now there is a bit of a mishmash of literature Review and results and not always clear what was found in this study.

The results section and discussion are now divided. We created results subheadings and we tried to create a more focused question/results/implications for each section of the results. We think these changes have improved the flow and clarity of the results and discussion.

Regarding their response to my comments:

Second response relating to community as a unit of analysis: This is still my biggest perceived weakness with the paper. Ok I buy the unit but still do not think it is sufficiently elaborated/theoretically justified in the text. What I mean is that I am looking for a deeper engagement with what community resilience is about. I understand that tax Revenue can be important, but as a devil's advocate I could argue that having 90% of fisher unemployed and 10% catching all the fish would be ok from that perspective, as long as they generate the same tax flow to the Community. but is this really true?

We have made additional efforts (please see below) to justify our use of the community as the relevant unit for assessing resilience (Lines 199-203). While it is true that ten percent fishing might generate the same direct tax revenues, however in small communities having many people employed generates other economic activity associated with the goods and services those people will purchase in their everyday lives.

Also, if by including a justification you mean the lines that appears in the last paragraph of the paper (line 235 as in the response) then I do not Think that is sufficient. it should come as you explain the justification of variables and design of the study.

We now include additional justification for the community as the relevant unit of analysis in introduction. We include how fishing brings tax revenue, commerce, and

often an identity to fishing communities. These are the core elements of the role fishing can play in community resilience (Lines 199-203).

Authors could do a better job of linking their choice of Community diversity to resilience - some suggested readings are: Chapter on diversity in the book Principles of Resilience that came out a few years ago, also "Management forcing increased specialization in a fishery system" J Hentati-Sundberg, J Hjelm, WJ Boonstra, H Österblom Ecosystems 18 (1), 45-61 - which gets into the interplay between individual diversity/specialization and system diversity.

We think our improvements mentioned above address this comment. We also discuss the differences between individual and community diversification, the challenges, and the opportunities (Lines 422-444).

You do not define turnover the first time it is used (line 65) which is still somewhat confusing - suggest moving the explanation up further.

We now define explicitly that turnover is the buying and selling of assets when it first comes up in text (Line 161).

Line 208-210- I dont understand what is meant?

We believe the reference is to “Alaska Native Corporations and economic development organizations.” We now include an additional description that these are organizations with the goals of improving the economic standing of native Alaskans and specific regions within the state (Lines 410-411).